# Setup and Characterization of a High-Throughput Luminescence-Based Serum Bactericidal Assay (L-SBA) to Determine Functionality of Human Sera against *Shigella flexneri*

**DOI:** 10.3390/biotech11030029

**Published:** 2022-07-27

**Authors:** Francesca Mancini, Francesca Micoli, Omar Rossi

**Affiliations:** GSK Vaccines Institute for Global Health s.r.l (GVGH), via Fiorentina 1, 53100 Siena, Italy; francesca.x.micoli@gsk.com (F.M.); omar.x.rossi@gsk.com (O.R.)

**Keywords:** serum bactericidal assay (SBA), *Shigella flexneri*, generalized modules for membrane antigens (GMMA), functional assay, human, luminescence, vaccine

## Abstract

Shigellosis represents a major public health problem worldwide. The morbidity of the disease, especially in children in developing countries, together with the increase of antimicrobial resistance make a vaccine against *Shigella* an urgent medical need. Several vaccines under development are targeting *Shigella* lipopolysaccharide (LPS), whose extreme diversity renders necessary the development of multivalent vaccines. Immunity against *Shigella* LPS can elicit antibodies capable of killing bacteria in a serotype-specific manner. Therefore, although a correlation of protection against shigellosis has not been established, demonstration of vaccine-elicited antibody bactericidal activity may provide one means of vaccine protection against *Shigella*. To facilitate *Shigella* vaccine development, we have set up a high-throughput serum bactericidal assay based on luminescence readout (L-SBA), which has been already used to determine the functionality of antibodies against *S. sonnei* in multiple clinical trials. Here we present the setup and intra-laboratory characterization of L-SBA against three epidemiologically relevant *Shigella flexneri* serotypes using human sera. We assessed the linearity, repeatability and reproducibility of the method, demonstrating high assay specificity to detect the activity of antibodies against each homologous strain without any heterologous aspecificity against species-related and non-species-related strains; this assay is ready to be used to determine bactericidal activity of clinical sera raised by multivalent vaccines and in sero-epidemiological studies.

## 1. Introduction

*Shigella* is a major contributor to diarrheal death worldwide [1,2,3], causing a range of diseases going from mild diarrhea to severe dysentery with fever. *Shigella* spp. account for 270 million diarrhea cases per year, with 90% occurring in low-and-middle-income countries (LMICs), and approximately 212,000 deaths in all age groups, of which 30% are in children under the age of 5 [1]. Though mortality rates from diarrheal diseases have experienced a decrease in the last decades, diarrhea morbidity remains, particularly in LMICs, impairing the health and quality of life of a huge number of underprivileged children. Additionally, *Shigella* can cause diarrhea among travelers and military recruits. The development of vaccines and therapeutics against *Shigella* is made a compelling priority by the preeminence of multidrug-resistant strains globally [4,5], and indeed the WHO’s Global Antimicrobial Resistance Surveillance System has identified *Shigella* as a priority pathogen for the development of new interventions [6].

*Shigella* consists of 4 species and more than 50 serotypes distinguished through the outer polysaccharide somatic O-antigen (OAg) of the lipopolysaccharide (LPS): *Shigella dysenteriae* (15 serotypes), *S. flexneri* (18 serotypes), *S. boydii* (20 serotypes), and *S. sonnei* (1 serotype) [7]. The prevalence of *Shigella* serotypes is extremely variable among countries and over time even within one region, thus posing a big challenge for vaccine development. Indeed, it is known that the immune responses to *Shigella* OAg are a key contributor to protection against infection [3,8]. The Global Enteric Multicentre Study (GEMS) showed that in sub-Saharan Africa and South Asia almost 90% of *Shigella* case isolates were represented by *S. flexneri* and *S. sonnei* [2].

No vaccines against *Shigella* are currently licensed, but several candidates have been developed and are being tested in preclinical and clinical settings [9]. Numerous strategies, including more traditional live attenuated or killed bacteria, subunit and glycoconjugate vaccines, as well as more innovative strategies such as bioconjugates and Generalized Modules for Membrane Antigens (GMMA) are actively being explored [10]. Although human and animal challenge trials with virulent *Shigella*, as well as observational studies in endemic areas, showed that the incidence of disease decreases following a prior *Shigella* infection [11], the development of vaccine candidates has been hindered by the lack of firm immunological correlates of protection, among other reasons, such as the need to develop multivalent vaccines able to protect against the highest number of serotypes as possible.

Based on evidence proving that vaccine-elicited functional antibodies represent serological correlates of protection against other bacterial pathogens [12], Shimanovich and co-authors have demonstrated that serum bactericidal assay (SBA) is a potential immune correlate of protection against shigellosis, since it provided an estimate of functional antibody activity associated with reduced shigellosis in humans [13]. To investigate functional antibody responses in humans, we established quantitative assays to measure *Shigella*-specific serum bactericidal antibody activity in sera from adult volunteers.

The working principle of SBA relies on reconstituting in vitro the conditions in which antibodies recognize antigen on the surface of target bacterium and bind to the exogenous complement, activating the classical complement pathway, thus resulting in bacteriolysis and the death of the target organism. The major issue with traditional SBA is that it relies on plating and counting the target bacteria. Therefore, conventional SBA has been often considered time-consuming and labor-intensive for the screening of large datasets and clinical samples [14]; however, efforts have been made in order to increase the analytical throughput of the assay, resulting in the development of both conventional [15], and non-conventional [16,17], high-throughput SBA. We have previously demonstrated the usefulness of a high-throughput SBA method based on luminescence (L-SBA) as a survival readout for several pathogens (including *S. flexneri* and *S. sonnei*, *Salmonella* Typhimurium, Enteritidis and Paratyphi A) using both animal [16,18], and human sera [18]. The number of viable bacterial cells surviving the complement-mediated antibody-dependent killing is quantified by measuring their metabolic ATP. After lysis of bacteria, ATP becomes available to trigger a luciferase-mediated reaction, resulting in a measurable luminescence signal. In L-SBA the level of luminescence detected is proportional to the number of living bacteria present in the assay wells, which is inversely proportional to the level of functional antibodies present in the serum [16]. The result of the assay is the IC50, the dilution of sera able to kill half of the bacteria present in the assay, thus representing the SBA titer of the sera. We have already demonstrated the possibility of using the L-SBA to determine the bactericidal activity of sera raised against *S. sonnei* GMMA in pre-clinical models [19], and clinical samples [20]. Indeed, we showed that antibodies elicited by *S. sonnei* monovalent GMMA-based vaccine-induced antibodies able to kill the bacteria in a complement-mediated manner both in European and US adults [21,22]. Here, we present the further development of this method for testing of clinical sera against *S. flexneri* 1b, 2a and 3a. We have characterized the method intra-laboratory by assessing its specificity, linearity, and precision. The developed assays will be used to determine the functionality of sera from adults intramuscularly immunized with a four-component *Shigella* vaccine that makes use of the GMMA technology [23], and could be easily expanded to further clinical studies or sero-epidemiological studies.

## 2. Materials and Methods

### 2.1. Bacterial Strains and Reagents

Working aliquots of *S. flexneri* 1b (Stansfield NTCT 5 strain), *S. flexneri* 2a (2457T strain) and *S. flexneri* 3a (6885 strain) stored frozen at −80 °C in 20% glycerol stocks were grown overnight (16–18 h) at 37 °C in a Luria Bertani (LB) medium, stirring at 180 rpm. The following day, the overnight bacterial growth was diluted in fresh LB medium in order to have an optical density at 600 nm (OD600) of 0.05 and incubated at 37 °C with 180 rpm agitation in an orbital shaker, until reaching OD600 of 0.22 ± 0.03. Baby (3- to 4-week-old) rabbit complement (BRC) was purchased from Cederlane, stored in 10 mL frozen aliquots, and thawed immediately prior to use. Phosphate Buffer Saline at pH 7 (PBS) was used for serum and bacteria dilutions. LPS was extracted from *S. sonnei* by hot phenol extraction as previously reported [24], whereas OAg was extracted from *S. flexneri* 1b, 2a and 3a and from *Salmonella* Typhimurium by direct acid hydrolysis, as previously reported [25]. All extracted polysaccharides were fully characterized in terms of sugar content, protein and nucleic acid impurities by a combination of analytical techniques, including High-Performance Liquid Size Exclusion Chromatography [25,26], micro-BCA and absorption at 260 nm as previously reported [19].

### 2.2. Serum Samples

The human serum tested (GVGH41148) was obtained by pooling high responder sera from volunteers originally enrolled in the H01_02TP clinical study (NCT01229176) performed in Pakistan and India, endemic areas for *Shigella*. High responders were selected based on their anti-*S. flexneri* 1b, 2a and 3a OAg IgG ELISA. Frozen 50 µL working aliquots of the serum were stored at −80 °C until use.

All tested samples were Heat Inactivated (HI) prior to testing in L-SBA at 56 °C for 30 min to remove endogenous complement activity. Various aliquots of HI GVGH41148 serum were used and treated as described below to determine the different assay parameters. Samples used to assess repeatability and intermediate precision: each sample consists of the same HI GVGH41148 serum; 12 identical samples were assayed each day by the same operator and the assay was repeated on three different days (36 samples in total, 12 on each day). Samples used to assess limit of detection (LoD) and limit of quantitation (LoQ): HI GVGH41148 was diluted 300 times (for L-SBA against *S. flexneri* 1b), 22 times (for L-SBA against *S. flexneri* 2a) or 84 times (for L-SBA against *S. flexneri* 3a) *v*:*v* in PBS to generate a sample with low but detectable SBA titer (expected IC50 to be around 100). Twelve identical HI GVGH41148 prediluted serum samples were assayed on the same day by the same operator starting from 1:4 dilution. Samples used to assess linearity: HI GVGH41148 serum was assayed neat or diluted 2, 4, 8, 16, 32-fold (*v*:*v*) with PBS prior to performing the assay (one IC50 obtained for each dilution). Samples used to assess specificity: two sets of samples were prepared to assess the homologous and heterologous specificity of the assay using HI GVGH41148 serum diluted 1:1 (*v*:*v*) in PBS alone or PBS supplemented with different quantity of homologous or heterologous purified polysaccharides. In the first experiment HI GVGH41148 serum was spiked with homologous (*S. flexneri* 1b, 2a or 3a) purified OAg at different final concentrations (4600, 2300, 1150, 500, 200, 50, 10, 1 µg/mL) and compared with a sample spiked 1:1 with PBS alone, incubated overnight (16–18 h) at 4 °C shaking at 200 rpm in an orbital shaker, prior to being tested. The lowest concentration of OAg between the ones tested able to inhibit the IC50 > 70% was then used in a second experiment to determine the heterologous specificity. In the second experiment, *S. sonnei* LPS, *S. flexneri* 2a and 3a OAg (for L-SBA against *S. flexneri* 1b), *S. flexneri* 1b and 3a OAg (for L-SBA against *S. flexneri* 2a), *S. flexneri* 1b and 2a OAg (for L-SBA against *S. flexneri* 3a) (heterologous but from the same species) and *Salmonella* Typhimurium OAg (heterologous from a different species) were used; internal controls for this experiment was represented by a sample preincubated overnight at 4 °C shaking at 200 rpm with an equal volume of PBS alone (undepleted) and a sample preincubated with homologous *S. flexneri* OAg (homologous). Each spiked sample was assayed on the same day.

### 2.3. Luminescent-SBA (L-SBA)

Serum bactericidal assay based on luminescent readout (L-SBA) was performed as previously described [20], with minor modifications. Briefly, different dilutions of HI test sera were incubated with bacteria in 96-well round bottom sterile plates (Corning, Glendale, AZ, USA)–the SBA plate-in the presence of exogenous complement (BRC). The HI sera were serially diluted in PBS (or LB for assay with *S. flexneri* 1b) in the SBA plate (25 µL/well) starting from 1:4 dilution (final dilution) for LoD and LoQ assessment and from 1:1000 (final dilution) in L-SBA against *S. flexneri* 1b, from 1:100 (final dilution) in L-SBA against *S. flexneri* 2a and from 1:300 (final dilution) in L-SBA against *S. flexneri* 3a for repeatability, intermediate precision, linearity and specificity assessment followed by 3-fold dilution steps up to 7 dilution points. One control well with no sera was added that represented control for non-specific complement killing as well as a sample diluted infinite-fold. Up to 12 different sera can be assayed within each SBA plate. Log-phase cultures (OD600 = 0.22 ± 0.02) were prepared as described above and diluted to approximately 1 × 10^6^ Colony Forming Unit (CFU)/mL in PBS. An adequate volume of reaction mixture containing the target bacterial cells (10 µL/well) and BRC (15 µL/well for assay with *S. flexneri* 1b and 3a and 7.5 µL/well for assay with *S. flexneri* 2a) as an external source of complement in assay buffer (57.5 µL/well of PBS for assay with *S. flexneri* 2a, 50 µL/well of PBS for assay with *S. flexneri* 3a and 50 µL/well of LB for assay with *S. flexneri* 1b) was prepared; 75 µL/well of the reaction mixture were added to each well of the SBA plate containing HI serum dilutions (final reaction volume 100 µL), mixed and incubated for 3 h at 37 °C. At the end of the incubation, the SBA plate was centrifuged at room temperature for 10 min at 4000× *g*. The supernatant was discarded to remove ATP derived from dead bacteria and SBA reagents. The remaining bacterial pellets were resuspended in PBS, transferred to a white 96-well plate (Greiner Bio-One, Roma, Italy) and mixed 1:1 *v*:*v* with BacTiter-Glo Reagent (Promega, Southampton, UK). The reaction was incubated for 5 min at room temperature on an orbital shaker at 600 rpm, and the luminescence signal was measured by a luminometer (Synergy HT, Biotek, Swindon, UK).

### 2.4. Calculations

The level of luminescence detected is directly proportional to the number of living bacteria present in the wells, which is inversely proportional to the level of functional antibodies present in the serum [16]. A 4-parameter non-linear regression was applied to the raw luminescence (no normalization of data was applied) obtained for all the sera dilutions tested for each serum; an arbitrary serum dilution of 10^15^ was assigned to the well containing no sera. Fitting was performed by weighting the data for the inverse of luminescence and constraining the curves to have a bottom between 0 and a defined threshold [20], calculated as follow. For *S. flexneri* 1b 600 counts per second (CPS) is the approximate value corresponding to the lowest luminescence detected at T180 for sera in all the wells in which bacteria are killed (400 CPS) plus the standard deviation (SD) of luminescence detected on those wells (155); for *S. flexneri* 2a 400 CPS is the approximate value corresponding to the lowest luminescence detected at T180 for sera in all the wells in which bacteria are killed (260 CPS) plus the SD of luminescence detected on those wells (140); for *S. flexneri* 3a 300 CPS is the approximate value corresponding to the lowest luminescence detected at T180 for sera in all the wells in which bacteria are killed (240 CPS) plus the SD of luminescence detected on those wells (60). To validate the assay plate, the average luminescence at T180 detected in wells with no sera had to be at least 5-fold for *S. flexneri* 1b, 3-fold for *S. flexneri* 2a and 2.5-fold for *S. flexneri* 3a higher with respect to luminescence detected at T0. To validate the dilution series, the highest luminescence detected in the dilution series at T180 had to be at least 0.6-fold the luminescence detected in the control well with no sera added. The results of the assay are expressed as the IC50 (the dilution of sera able to kill half of the bacteria present in the assay), represented by the reciprocal serum dilution that results in a 50% reduction of luminescence (and thus raising 50% growth inhibition) respect to the control represented by the wells containing no sera. GraphPad Prism 7 software (GraphPad Software, La Jolla, CA, USA) was used for fitting and IC50 determination.

### 2.5. Statistical Analysis

Statistical analyses were performed with Minitab 18 (Minitab Inc., Chicago, IL, USA) as described in the results section. ANOVA with variance component analysis (mixed effect model with random factors) was used to estimate the intermediate precision (defined as the variability among different days and different operators), the repeatability (defined as the variability under the same operating conditions over a short interval of time), and to evaluate the contributions of the day of analysis to the variability. Regression analysis was used to evaluate linearity.

### 2.6. Ethical Statement

The human serum pool used in this study was derived from subjects enrolled in the clinical trial registered with ClinicalTrials.gov number NCT01229176. The relevant ethical and regulatory approval was obtained from the respective institutional and national ethics review committees. Written informed consent was obtained before enrollment from the subjects and the trial was designed and conducted in accordance with the Good Clinical Practice Guidelines and the Declaration of Helsinki.

## 3. Results

### 3.1. Optimization of L-SBA on Human Sera against S. flexneri 1b, 2a and 3a Strains

In order to determine the possibility of assessing the serum bactericidal activity of human sera against *S. flexneri* 1b, 2a and 3a by L-SBA, we tested an anti-*S. flexneri* IgG human standard serum, GVGH41148. HI GVGH41148 was used to setup the assay conditions and characterize the assay prior moving on with testing the functionality of clinical samples. Initial experiments were conducted to test the behavior in L-SBA of GVGH41148 under experimental conditions (% of baby rabbit complement and target bacterial strain) already established with pre-clinical sera [27,28]. Experiments were conducted in comparison to mouse standard sera extensively tested in pre-clinical studies [16], thus serving also as bridging. Assay conditions developed in pre-clinical studies were found to be optimal also when using human sera, confirming the non-specific killing of BRC at the selected percentage (15% for *S. flexneri* 1b and 3a and 7.5% for *S. flexneri* 2a) in the assay (Figure 1). We had already verified that human pre-immune sera gave no aspecific killing [18]. A prozone effect was detected at 1:4 dilution when testing GVGH41148 against *S. flexneri* 3a, up to 1:12 dilution when testing GVGH41148 against *S. flexneri* 2a and up to 1:324 dilution when testing GVGH41148 against *S. flexneri* 1b. For a curve readout vs. dilution (in this case luminescence vs. serum dilution) it is defined prozone effect, a condition in which for the least diluted points tested the readout value (luminescence) is higher than that obtained with points more diluted; however, HI GVGH41148 has been tested in following experiments starting from 1:1000 (final dilution) in L-SBA against *S. flexneri* 1b, from 1:100 (final dilution) in L-SBA against *S. flexneri* 2a and from 1:300 (final dilution) in L-SBA against *S. flexneri* 3a. With the identified assay conditions, L-SBA was characterized for each serotype by assessing precision (both in terms of repeatability and intermediate precision), specificity, linearity, as well as determining limit of detection and quantitation of the assays.

### 3.2. Precision

The precision of the method expresses the closeness of agreement among multiple analyses of the same sample tested under the prescribed conditions. Both repeatability (intra-assay variation) and reproducibility as intermediate precision (inter-assay variation) were evaluated. Thus, 12 replicates of HI GVGH41148 serum were assayed on the same day and repeated on three different days (36 measurements in total, Figure 2). Log-transformed IC50 results from the three runs were analyzed with Minitab 18 (Minitab, LLC, State College, PA, USA) applying a Mixed Effects Model considering the day as random factor in order to determine the repeatability (CV% R) and intermediate precision (CV% IP) of the assay. The assay against *S. flexneri* 1b was characterized by an intermediate precision (CV% IP) of 5.37% and a repeatability (CV% R) of 3.17%; the assay against *S. flexneri* 2a was characterized by an intermediate precision (CV% IP) of 1.57% and a repeatability (CV% R) of 1.55%; the assay against *S. flexneri* 3a was characterized by an intermediate precision (CV% IP) of 3.30% and a repeatability (CV% R) of 1.47%. For the three assays, the variance of the reproducibility was not significant.

Average LogIC50 from all measurements, standard deviation and coefficient of variation (CV%) resulted to be those reported in Table 1.

### 3.3. Linearity

To assess the linearity of the assays, HI GVGH41148 serum was pre-diluted in PBS (neat, 2-fold, 4-fold, 8-fold, 16-fold and 32-fold times, respectively) before being probed against the three *S. flexneri* serotypes in L-SBA. Each pre-dilution was prepared independently.

We considered the average of Log (IC50) of the undiluted serum obtained when assessing the precision of the assay (Table 1) as the “true value”, and from this one we calculated the expected Log (IC50) based on the dilutions by volume performed (IC50 theoretical). A regression analysis was performed on Log (IC50 experimentally obtained) vs. Log (IC50 theoretical) (Figure 3).

From the analyses of variance, the linear models were significant (*p*-value < 0.01) and the residuals of the linear regression models were normally distributed; for each serotype, the intercept (Constant in Table 2) was not significantly different from zero (*p*-value > 0.05), and the slope (T term in Table 2) was not significantly different from 1 (95% CI ranges included 1).

### 3.4. Specificity

The specificity of an analytical procedure is its ability to determine solely the concentration of the analyte that it intends to measure. In case of the L-SBA against *S. flexneri* 1b, 2a and 3a is the *S. flexneri* 1b, 2a and 3a OAg portion of the LPS, respectively.

We initially assessed the homologous specificity by spiking homologous *S. flexneri* OAg to HI GVGH41148 at the final concentrations of 4600, 2300, 1150, 500, 200, 50, 10, 1 µg/mL in PBS prior to being assayed in L-SBA against *S. flexneri* 1b, at the final concentrations of 1000, 500, 200, 50, 10, 1 µg/mL in PBS prior being assayed in L-SBA against *S. flexneri* 2a, at the final concentrations of 5400, 2700, 1400, 500, 200, 50, 10, 1 µg/mL in PBS prior being assayed in L-SBA against *S. flexneri* 3a. The undepleted control was represented by HI GVGH41148 serum incubated with an equal volume of PBS alone. The percentage of inhibition was determined by calculating the decrease in the observed SBA titer between samples pre-treated with a competitor and undepleted control. The lowest homologous OAg concentration, among the ones tested, that could cause a reduction of the IC50 ≥ 70% compared to the undepleted control sample is was selected to assess the heterologous specificity (when feasible). Specifically, 4.6 mg/mL homologous OAg concentration was selected for *S. flexneri* 1b, 0.5 mg/mL homologous OAg concentration was selected for *S. flexneri* 2a, and 5.4 mg/mL homologous OAg concentration was selected for *S. flexneri* 3a.

A second set of experiments was performed to determine the heterologous specificity; this was carried out by pre-incubating HI GVGH41148 serum with an equal volume of heterologous competitors (heterologous *S. flexneri* OAg, *S. sonnei* LPS and *S.* Typhimurium OAg) at the final concentration identified for assessing the homologous specificity. When the same concentration of homologous competitor could not be used for the heterologous competitors, the highest possible concentration was used. Internal controls for this experiment were represented by a sample preincubated with an equal volume of PBS alone (undepleted), and by a sample preincubated with homologous *S. flexneri* OAg (to confirm the homologous specificity). Specificity was determined as % IC50 inhibition; calculated using the following formula:%IC50 inhibition = (IC50 of the undepleted sample) − (IC50 of the sample  pre-treated with competitor)/(IC50 of the undepleted sample) × 100

Depletion with homologous competitor caused an inhibition of IC50 of at least 94% for all the three assays, confirming their high specificity, whereas depletion with heterologous antigens resulted in an absent or a marginal (<6%) decrease in SBA titer, suggesting the absence of any non-*S. flexneri* polysaccharide-specific response in the assay (Figure 4).

### 3.5. Limit of Detection (LoD) and Limit of Quantitation (LoQ)

Finally, we determined the Limit of Detection (LoD) and the Limit of Quantitation (LoQ) of the assays, representing the lowest SBA titer than can be detected under the assay conditions and the lowest SBA titer that can be quantified with a suitable precision, respectively. To do so, HI GVGH41148 was pre-diluted in PBS to generate a sample with low but detectable SBA titer (aiming to determine limits in the worst-case scenario possible for each assay, and thus the one expected to give the highest variability). Thus, GVGH41148 was pre-diluted 300-fold prior being assayed in L-SBA against *S. flexneri* 1b, 22-fold prior to being assayed in L-SBA against *S. flexneri* 2a and 84-fold prior being assayed in L-SBA against *S. flexneri* 3a.

SBA titers (LogIC50) were obtained for each sample from Luminescence versus Log sera dilutions fitting (Figure 5).

Calculations of LoD and LoQ were performed accordingly to the ICH guideline Q2(R1) [29], by using the standard deviation (SD) of Log transformed SBA titers obtained for the samples and the lowest serum concentration tested in the assay (X, in our case = 4) according to the following formulas: LoD = 10^(3.3 × SD) × X and LoQ = 10^(10 × SD) × X. LoD and LoQ resulted to be those reported in Table 3.

## 4. Discussion

A clear correlate of protection against shigellosis has not yet been defined, however epidemiological studies and challenge/rechallenge experiments in humans and nonhuman primates have shown that protection against *Shigella* is serotype specific [30,31]. Serotype specificity is determined by the OAg of LPS, therefore immune responses targeting the OAg are likely responsible for protection against shigellosis. Although the OAg-specific immune response has traditionally been quantified by antigen-specific IgG detection through ELISA, presumably a functional readout, such as opsonophagocytosis and/or complement activation, can better determine the effectiveness of antibodies raised upon vaccination [32]. A recent study reported a strong association between *S. flexneri* 2a-specific SBA titers in human adult volunteers and reduced clinical disease post-challenge with wild-type organisms [13], thus supporting the value of this assay to potentially predict vaccine efficacy. Also in Clarkson et al. [33], vaccinees protected against *S. flexneri* 2a infection had higher SBA titers compared to cases; moreover, Nahm et al. reported that, although a clear immune correlate of protection has not been established, antibodies with bactericidal capacity may provide one means of protecting against shigellosis underling the importance of measuring the functional capacity of antibodies, as opposed to only binding activity [15]. Furthermore, bactericidal activities have been detected in sera from infected individuals living in regions of *Shigella* endemicity [34,35,36,37]. So far traditional complement-mediated killing assays have been used to assess antibodies bactericidal activity against *Shigella* [15]. These assays, however, are time consuming and operator dependent, therefore results can be highly variable and difficult to compare among different labs. Hence, we have previously developed an SBA based on luminescence (L-SBA) which represents a simple, high-throughput assay to measure serum bactericidal activity against several Gram-negative bacteria [16]. L-SBA was setup and characterized for testing of human sera against *S. sonnei* [20], successfully used to prove the complement-mediated serum bactericidal activity of antibodies elicited by the *S. sonnei* GMMA vaccine in adults enrolled in a Phase 1 dose escalation trial followed by a booster extension [22], and in US adults [21].

In this study, we have presented the optimization and characterization of L-SBA to detect bactericidal activity of human sera against three epidemiologically relevant *S. flexneri* strains: *S. flexneri* 1b, 2a and 3a. The assay will be used to assess the bactericidal activity of antibodies induced by the 4-component GMMA-based vaccine currently being tested in a Ph1/2 clinical trial [23], and can be used to better characterize the immune response elicited by other vaccines in development. The assay, firstly developed for *S. sonnei* [20], has been easily extended here to the other three *S. flexneri* serotypes. Based on this we expect that it could be rapidly adapted to additional Shigella serotypes [20,23], antibodies; moreover, the high-throughput L-SBA method can be applied to investigate in depth the immune response to several *Shigella* serotypes in adults and children naturally exposed to these bacteria allowing to determine if there is a correlation between anti-OAg IgG and SBA responses in different age populations in endemic countries.

Here human L-SBA against *S. flexneri* 1b, 2a and 3a have been fully characterized, showing that in the working conditions tested, the three assays are able to detect sera having a SBA titer as little as 7, 6, and 11, respectively, with ideally no upper limit of detection; moreover, the three L-SBA against *S. flexneri* 1b, 2a and 3a can quantify with precision sera with IC50 of 22, 13, and 83, respectively. The assays showed very low variability, both in terms of repeatability (intra-assay variation) and reproducibility as intermediate precision (inter-assay variation), which were estimated to be below 5.5% for all the assays. Day of analysis did not contribute to the overall variability. Furthermore, L-SBA was specific for the target active ingredient of the vaccine candidate, the OAg portion of the LPS. Indeed, by depleting the serum with homologous OAg, at least a 94% reduction of IC50 was observed, whereas no depletion was detected when depleting sera with heterologous *S. flexneri* OAg, *S. sonnei* LPS and *S*. Typhimurium OAg. The linearity of the assay was also assessed, and it was found to be good within the tested range.

The L-SBA assay allows the testing of 11 serum samples in each plate, plus a standard serum to validate each plate. Since up to 12 plates can be run in a day by a single operator, the assay has a high-throughput; moreover, our assay uses standard reagents and requires only a luminometer to detect ATP, making it simple enough to be adopted by any laboratory worldwide. We have also demonstrated the possibility to further increase the throughput of our L-SBA method by performing the assay in 384-well format [38], and thus the possibility to further increase the already high-throughput if necessary, especially for large clinical or sero-epidemiological studies.

In conclusion, L-SBA applied to human sera has already proved to be suitable to perform clinical analysis in high-throughput [22]. Here we have extended the optimization and characterization of the assay to *Shigella flexneri* serotypes in order to facilitate the development of *Shigella* vaccines in multi-component formulations.

## Figures and Tables

**Figure 1 biotech-11-00029-f001:**
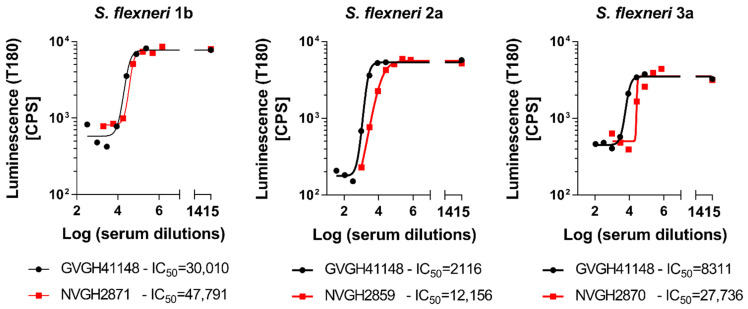
Comparison of pre-clinical L-SBA standard sera (NVGH2871, NVGH2859 and NVGH2870) and human standard serum (GVGH41148) in L-SBA against *S. flexneri* 1b, 2a and 3a. 4PL fitting of luminescence at T180 (CPS) vs. log-transformed serum dilution. In the graphs, IC50 was obtained by testing mouse standard serum NVGH2871 (for *S. flexneri* 1b), NVGH2859 (for *S. flexneri* 2a), NVGH2870 (for *S. flexneri* 3a), and GVGH41148 (human serum) are reported in open squares and filled circles, respectively.

**Figure 2 biotech-11-00029-f002:**
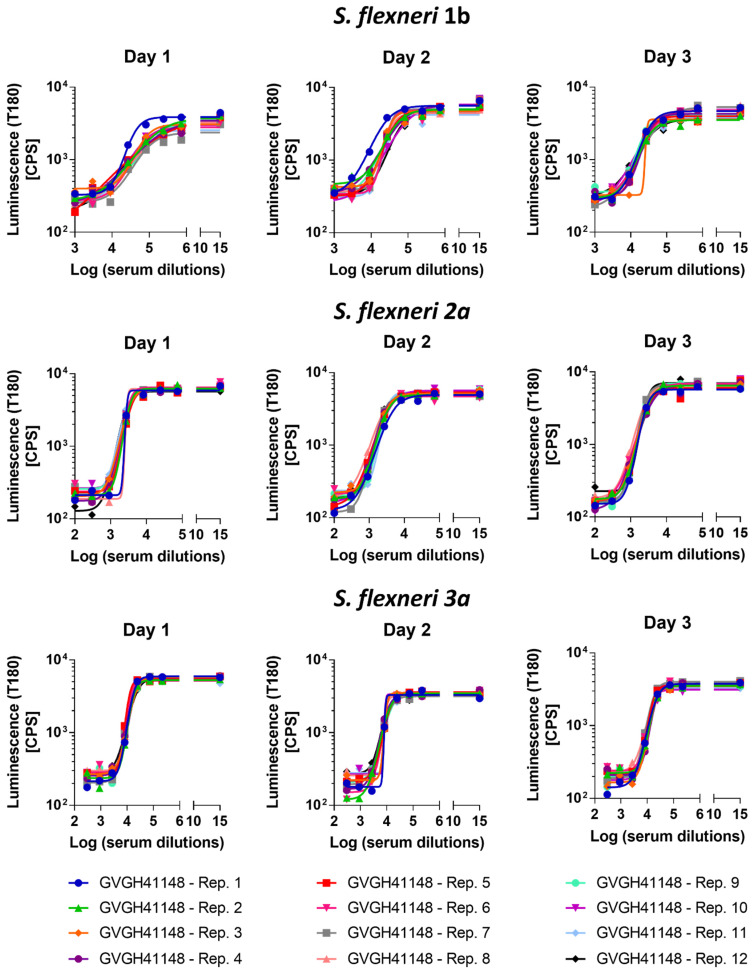
Luminescence vs. Log serum dilution tested to determine the precision of the assay. Symbols represent single measurements. Solid lines represent the fitted 4 parameters curves to single measurements data for each sample assayed for the three *S. flexneri* serotypes tested.

**Figure 3 biotech-11-00029-f003:**
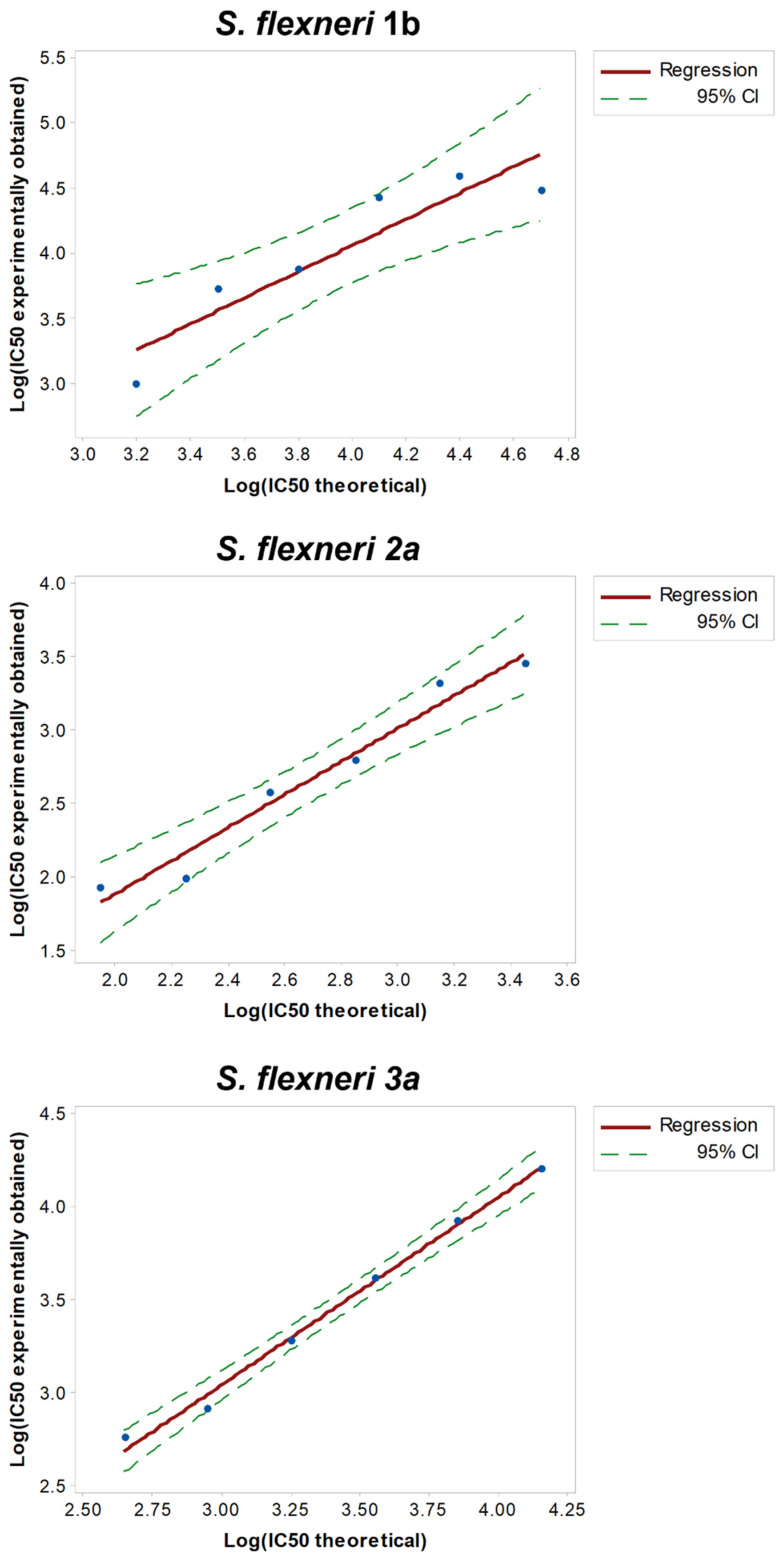
Linearity. Log (IC50 theoretical) obtained for each sample versus Log (IC50 experimentally obtained). Single datapoints are indicated with blue dots. The red solid line represents the linear regression and the green dashed lines the 95% confidence interval (CI).

**Figure 4 biotech-11-00029-f004:**
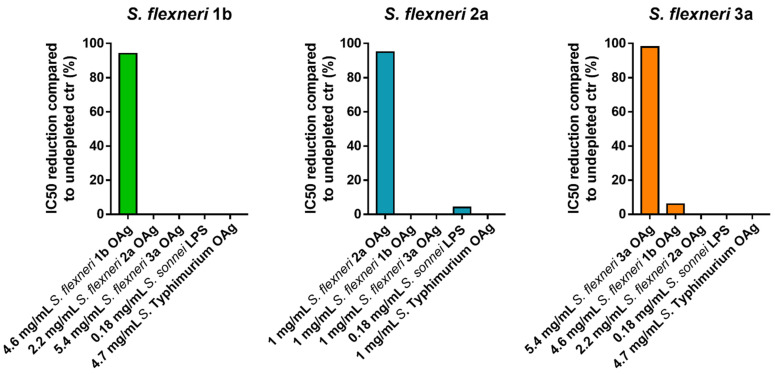
Specificity. IC50 reduction compared to undepleted control samples after incubation with homologous or heterologous competitors.

**Figure 5 biotech-11-00029-f005:**
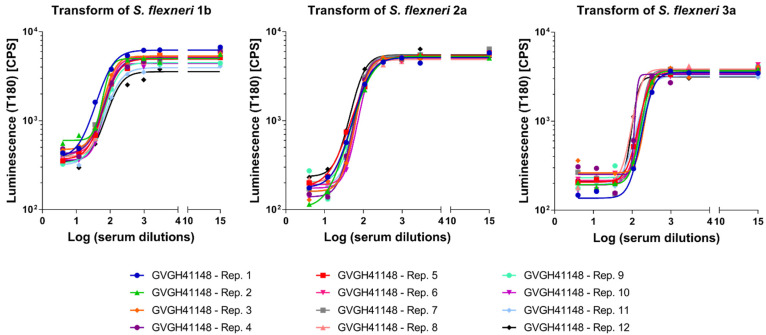
Luminescence vs. Log serum dilution of samples tested to assess LoD and LoQ. Symbols represent single measurements. Solid lines represent the fitted 4 parameters curves to single measurements data for each sample assayed against the three *S. flexneri* serotypes.

**Table 1 biotech-11-00029-t001:** Average LogIC50, SD and CV% of GVGH41148 serum against each of the three *S. flexneri* serotypes tested.

	*S. flexneri* 1b	*S. flexneri* 2a	*S. flexneri* 3a
Average Log (IC50)	4.65	3.45	4.13
SD	0.22	0.05	0.12
CV%	4.76	1.56	2.85

**Table 2 biotech-11-00029-t002:** Coefficients of the regression analysis for each of the three *S. flexneri* serotypes tested. The Constant term represents the intercept of the regression line whereas the T term represents the slope of the regression line.

	Term	Coef	SE Coef	95% CI	*p*-Value
***S. flexneri* 1b**	**Constant**	0.041	0.802	(−2.184, 2.267)	**0.961**
	**T**	1.005	0.201	**(0.446, 1.563)**	
***S. flexneri* 2a**	**Constant**	−0.362	0.293	(−1.176, 0.451)	**0.284**
	**T**	1.123	0.107	**(0.827, 1.419)**	
***S. flexneri* 3a**	**Constant**	0.012	0.152	(−0.410, 0.435)	**0.939**
	**T**	1.0087	0.0442	**(0.8859, 1.1315)**	

**Table 3 biotech-11-00029-t003:** LoD and LoQ for each of the three *S. flexneri* serotypes tested.

	*S. flexneri* 1b	*S. flexneri* 2a	*S. flexneri* 3a
**LOD**	7	6	10.9
**LOQ**	21.8	13.4	82.6

## Data Availability

Data are contained within the article.

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
