# Peer review of "Setup and Characterization of a High-Throughput Luminescence-Based Serum Bactericidal Assay (L-SBA) to Determine Functionality of Human Sera against Shigella flexneri"

_biotech, 2022, doi:10.3390/biotech11030029_

Round 1
Reviewer 1 Report
In this work, Mancini et al present a method for fighting Shigellosis with the optimization and characterization of L-SBA to detect bactericidal activity of human sera against three epidemiologically relevant S. 441 flexneri strains.
My comment is accepted in this form.
Author Response
We really appreciate the positive comments. Thank you very much.
Reviewer 2 Report
This paper was written to analyze the function determination of human serum for Shigella flexneri, which is a problem in some developing countries. Setup and characterization of a High-Throughput L-SBA is well documented. I think that there is no problem in publishing if only simple typos and sentence errors are reviewed.
However, the figure needs to be partially modified.
1. In Figure 1, the difference in Luminescence (T180) CPS according to the log (serum dilutions) is not clearly shown because the dots in the graph are too large.
2. In Figures 2 and 5, it is necessary to make the graph slightly larger to check the luminescence (T180) value.
Author Response
We appreciate the positive and constructive comments.
The manuscript has been carefully revised to edit typos and sentence errors.
We have also improved the quality of Figure 1, 2 and 5 to make them more clear.
Reviewer 3 Report
I admit that the subject of the review does not perfectly coincide with my field of study and research. Nonetheless, I have evaluated the article as carefully as possible. The article has been set up very well and with very valid motivational bases. The results obtained seem to me to be very valid and useful for a future application in the field of vaccinations for humans. I found only small typos in the text: I reported them in the file that I am attaching to my review.

Author Response
We appreciate the very positive comments and careful editing. We have modified the manuscript according to reviewer's suggestions and comments.